# Experimental Study on Hybrid Effect Evaluation of Fiber Reinforced Concrete Subjected to Drop Weight Impacts

**DOI:** 10.3390/ma11122563

**Published:** 2018-12-17

**Authors:** Jun Feng, Weiwei Sun, Hongzhou Zhai, Lei Wang, Haolin Dong, Qi Wu

**Affiliations:** 1National Key Laboratory of Transient Physics, Nanjing University of Science and Technology, Nanjing 210094, China; 2Institute of Industrial Science, The University of Tokyo, Tokyo 1538505, Japan; zhaihongzhou@nuaa.edu.cn (H.Z.); wuqi@iis.u-tokyo.ac.jp (Q.W.); 3Department of Civil Engineering, Nanjing University of Science and Technology, Nanjing 210094, China; lei_wangnjust@163.com (L.W.); 117113022265@njust.edu.cn (H.D.); 4State Key Laboratory of Mechanics and Control of Mechanical Structures, Nanjing University of Aeronautics and Astronautics, Nanjing 210016, China

**Keywords:** hybrid fiber reinforced concrete, drop weight test, impact energy, hybrid effect index

## Abstract

In this paper, the impact energy potential of hybrid fiber reinforced concrete (HFRC) was explored with different fiber mixes manufactured for comparative analyses of hybridization. The uniaxial compression and 3-point bending tests were conducted to determine the compressive strength and flexural strength. The experimental results imply that the steel fiber outperforms the polypropylene fiber and polyvinyl alcohol fiber in improving compressive and flexural strength. The sequent repeated drop weight impact tests for each mixture concrete specimens were performed to study the effect of hybrid fiber reinforcement on the impact energy. It is suggested that the steel fiber incorporation goes moderately ahead of the polypropylene or polyvinyl alcohol fiber reinforcement in terms of the impact energy improvement. Moreover, the impact toughness of steel-polypropylene hybrid fiber reinforced concrete as well as steel-polyvinyl alcohol hybrid fiber reinforced concrete was studied to relate failure and first crack strength by best fitting. The impact toughness is significantly improved due to the positive hybrid effect of steel fiber and polymer fiber incorporated in concrete. Finally, the hybrid effect index is introduced to quantitatively evaluate the hybrid fiber reinforcement effect on the impact energy improvement. When steel fiber content exceeds polyvinyl alcohol fiber content, the corresponding impact energy is found to be simply sum of steel fiber reinforced concrete and polyvinyl alcohol fiber reinforced concrete.

## 1. Introduction

Concrete is one of the most widely used construction materials, but its tensile strength is relatively low in contrast with the compressive strength. This defect limits the further application of concrete in construction and building [1,2]. Using emerging materials with improved mechanical, dynamic and durability properties, fiber reinforced concrete (FRC) is ideal for disaster prevention and mitigation of explosion/impact applications where high impact resistance and energy absorption capacities are required [3]. Fiber also helps arrest micro cracks before the peak, therefore, during the hardening behavior of the composite the cracking are initiated to enhances the post-cracking behaviour due to improved stress transfer provided by the fibers bridging the cracked sections [4,5,6]. The fibers used are mainly steel fibers, carbon fibers, polymer fibers and natural fibers (e.g., the flax fibers) [7,8,9]. Among the polymer fibers, polypropylene (PP) and polyvinyl alcohol (PVA) fibers have attracted most attention due to the outstanding toughness of concrete reinforced with them [10,11,12,13]. Since, concrete is a complex material with multiple phases which include large amount of C-S-H gel in micron-scale size, sands in millimeter-scale size, and even coarse aggregates in centimeter-scale size. Therefore, the properties of FRC will be improved in certain level, but not whole levels if reinforced only by one type of fiber [14]. For instance, steel fibers are supposed to strengthen concrete in the sacle of coarse aggregate, PP or PVA fibers are suitable for the fine aggregate scale crack prevention and carbon nanotubes are proven to improve the strength in the scale cement grains [15].

To provide hybrid reinforcement, in which one type of fiber is smaller, so that it bridges microcracks of which growth can be controlled. This leads to a higher tensile strength of the composite. The second type of fiber is larger, so that it can arrest the propagating macrocracks and can substantially improve the toughness of the composite [16]. In practice, using hybridization with two different fibers incorporated in a common cement matrix, the hybrid fiber reinforced concrete (HFRC) can offer more attractive engineering properties because the presence of one fiber enables the more efficient use of the potential properties of the other fiber [7,17,18,19]. In addition, HFRC shows improved structural behaviour when compared with conventional concrete, qualities such as less spalling and scabbing under impact loadings [20,21,22,23].

There are several test methods that evaluate the impact strength of FRC where the simplest method is the drop-weight test proposed by the ACI (American Concrete Institution) committee 544 [24]. Nia et al. [25] investigated the increase of first crack initiation and final fracture impact strength of FRC with respect to the plain concrete. Sivakumar and Santhanam [17] compared the FRC with metallic and non-metallic fibers and found that the steel fiber generally plays an important role in the energy absorbing mechanism (bridging action), whereas non-metallic fiber could delay the formation of the micro-cracks. The statistical analysis of impact strength of steel-polypropylene hybrid fiber reinforced concrete (Steel-PP HFRC) carried out by Song et al. [26] with drop-weight tests, demonstrated that the HFRC provides higher improvement on the reliabilities of the first-crack strength and failure strength than the steel fiber reinforced cementitious composites. Conducting repeated drop-weight tests, Yildirim et al. [27] concluded that the steel fiber reinforcement as well as steel-polypropylene hybrid reinforcement can significantly improve the impact performance of concrete. It was found by Wille et al. [28] that there is no significant difference among various types of fibers, such as smooth, hooked or twisted opposite to fiber volume fraction which brings significant difference in terms of tensile strength and energy absorption capacity of resulting material. Providing reasonable trade-off between workability and mechanical properties of the mixture, straight steel fiber was chosen in this study to improve ductility of the composite.

The objective of this work is to investigate the hybrid effect of different fibers on the impact toughness under drop weight tests. As common and popular types of fiber, steel, polypropylene and polyvinyl alcohol fibers were chosen to produce the HFRC. The uniaxial compression and 3-point bending tests (3PBT) were performed to determine the compressive and flexural strength. Drop weight tests were further conducted to compare the fiber reinforcement effect whereby the impact energy was evaluated by first crack and failure strength. A comprehensive analysis was carried out to evaluate the hybrid effect of steel fiber reinforced with PP or PVA fiber on the repeated drop weight impact responses. The experimental results may provide an effective way to improve the impact toughness of fiber reinforced concrete material and structures.

## 2. Experimental Programme

The fiber content effect on FRC mechanical and mixing contents was experimentally studied by Yoo et al. [29], indicating that 2 vol.% fibers provides the best performance in fiber pullout behavior including average/equivalent bond strength and pullout energy. Therefore, this work chooses to investigate the HFRC with about 2% content fiber by volume.

### 2.1. Material Composition

The details of the concrete mixture proportions in this study are normalized and listed in Table 1. Portland cement (P.I 42.5) was used herein as a cementitious material and fly ash was added as a mineral active fine admixture. Ground fine quartz sand worked as fine aggregate and its gradation curve is plotted in Figure 1. The water-binder ratio and sand-binder ratio were 0.25 and 0.45, respectively. To improve fluidity, a high performance water-reducing agent, ploycarboxylate superplasticizer (DC-WR2) was also added which may contribute to the self-compacting property. In this experimental study, polypropylene, polyvinyl alcohol and steel fibers used for ultra-high-performance hybrid fiber reinforced concrete (UHP-HFRC) reinforcement were comparatively depicted in Figure 2. The geometric information and mechanical properties of these three fibers are listed in Table 2. It is demonstrated that the steel fiber is stronger and stiffer, while the PP fiber is finer and more flexible and ductile. To investigate the hybridization of PP, PVA and steel fiber (SF) reinforcement effect on HFRC impact energy, 16 mixtures with a single type of fiber reinforcement or hybrid fiber reinforcements at a total content of 1.5–2.5%, by volume of the concrete, are produced for further studied.

### 2.2. Mix Proportioning and Concrete Production

The mixing procedure of FRC needs to be rigorously controlled to ensure the resulting matrix with good workability, particle distribution and compaction. Noting that the small particles tend to agglomerate which may break the chunks when the particles are dry. It is suggested to blend all fine dry particles before adding water and superplasticizer. In the climatic chamber with 90% humidity, the FRC samples were prepared with the following mixing procedures. Firstly, the dry cementitious materials (cement, fly ash) and quartz sand were put together simultaneously and mixed for 1 min at a low speed to achieve the binder-sand mixture. Afterwards, the water and superplasticizer were mixed and gradually poured into the mixture to improve its flowability. Finally, the fibers were slowly added and mixed for another 5 to 8 min to ensure that all the fibers were evenly distributed in the mortar. 24 h later, the specimens were removed from moulds and cured for another 6 and 27 days at room temperature with humidity >95%.

The self-compactability of the fresh mixtures was qualitative evaluated since the FRC mixtures exhibit excellent deformability and proper stability to flow under its own weight. Furthermore, it was observed that mixtures with higher PVA or PP show poorer flowability because more porous microstructure might be generated due to relatively poor consolidation condition compared to steel fiber case.

### 2.3. Test Method

With the foregoing concrete samples reparation procedure, the quasistatic tests, including uniaxial compression (UC) and 3-point bending, were performed to investigate the effect of fiber reinforcement on the compressive and flexural strength. It worth noting that since only fine gradation of quartz sand were used as aggregate, we prepare the UC and 3PBT specimens with similar sizes adopted in [30,31,32]. Moreover, the hybrid effect of steel fiber and polymer (PP and PVA) fiber on the impact performance of the HFRC via the drop weight test. In this section, the experimental programme are explained in detail. Then experimental results will be reported and discussed based on the average values of tests for 3 specimens.

Specimens of 40 mm × 40 mm × 40 mm were cast for quasi-static compressive strength testing. Three samples of each mix were tested to determine the uniaxial compressive strength. Abrasive paper was used to smooth the surface of the specimens. The non-casting surfaces of the cube specimen were used as bottom and top surfaces of the compression test to ensure complete contact with the platen of the universal testing machine in Figure 3a. A loading rate of 2.4 kN/s was adopted to conduct the uniaxial compression test.

In order to analyze the fiber effect on the flexural strength of FRC, 3-point bending tests were conducted herein with specimens of different fiber mixes. The dimension of the tested beams are 40 mm (width *b*) × 40 (depth *d*) mm in cross-section, and 160 mm in total length where the span *l* is fixed at 120 mm. To insure quasi-static condition, the 3PBT was conducted at a rate of 0.5 mm/min for load cell of MTS machine. In Figure 3b, the beam was put on the rolling supports which can be deemed as fixed vertical constrain. During the bending test, the displacement and the corresponding load value were recorded. Related to the peak load $F_{P}$, span *l*, depth *d* and width *b*, the nominal flexural strength $f_{f}$ is expressed as $f_{f}=3F_{P}l/\left(2bd^{2}\right)$ [33].

The impact test was carried out in accordance to ACI Committee 544 drop weight impact test [24]. The test procedure is as follows: a repeatedly dropping hammer with 10.26 kg mass was released from a height of 457 mm. The hammer hit a 63.5 mm diameter hardened steel ball which was fixed at the center of the top surface of the concrete specimen. The hammer dropped to impact on the steel ball which transferred the pulse with the contact. The test apparatus, test set up and dimensions are depicted in Figure 4. In the drop weight impact test, the number of blows to cause the first visible crack was recorded as the first crack strength ($N{}_{1}$) while the failure strength ($N{}_{2}$) was defined as the number of blows to spread the cracks sufficiently (complete fracture), i.e. the concrete species touched three of the steel lugs [34].

With reference to [19,35], the impact toughness is defined as the impact energy absorbed by the concrete transformed from the drop hammer potential energy during the drop weight impact tests. Thus, the calculation of impact toughness, namely impact energy, is written as follows:(1)E=N·mgh
where *E*, *N*, *m*, *g*, *h* denote impact energy (impact toughness), number of the repeated impact times that the first visible crack and the final failure occur, mass of the drop hammer, gravitational acceleration, drop height of fall, respectively.

## 3. Test Results and Discussion

After performing uniaxial compression, 3-point bending and drop weight impact tests, PC and FRC with different types fibers are comparatively studied herein.

### 3.1. UC and 3PBT Results

As listed in Table 3, the averages of three samples of 17 mixtures of concrete for 7-day and 28-day compressive strengths (7-d $f_{c}$ and 28-d $f_{c}$) as well as nominal flexural strength (7-d $f_{f}$ and 28-d $f_{f}$) are summarized in detail. The mix numbers are featured with ‘S’ denoting steel fiber, ‘P’ representing PP fiber, ‘A’ meaning PVA fiber and their subscript values are the volume fraction percentage for corresponding fiber. For instance, S${}_{0.5}$P${}_{1.0}$ is the HFRC with 0.5% steel fiber + 1.0% PP fiber. The 7-d $f_{c}$ and 28-d $f_{c}$ of the UHP-HFRC with only PP, PVA and steel fiber are 90.10, 84.81 and 115.66 MPa, respectively. It worth noting that ploymer microfibers at high volume content negatively affect the flowability of mixture due to high specific surface area. The lower strength of PP or PVA incorporated mixtures are usually attributed to the higher porosity, supporting the finding that the finest fiber (PVA) gives the lowest strength values at 2% usage dosage. For 2% hybrid fiber reinforcement, both the 7-d $f_{c}$ and 28-d $f_{c}$ increase as steel fiber content increases. Although PP and PVA fibers are not expected to increase the compressive strength [36], it is observed that polyvinyl alcohol fiber could better improve the compressive strength than polypropylene fiber when the SF content remains constant meanwhile the flexural strength can be better improved by PP fiber incorporation, comparing Steel-PP HFRC (mix No. S${}_{0.5}$P${}_{1.0}$ to S${}_{1.5}$P${}_{0.5}$) with Steel-PVA HFRC (mix No. S${}_{0.5}$A${}_{1.0}$ to S${}_{1.5}$A${}_{0.5}$).

The flexural responses of 3PBT are presented in Figure 5. The addition of mono PP and steel fiber can considerably improve the post-crack behaviour in Figure 5a suggesting the residual strength and toughness of FRC beam are obviously enhanced compared to the PC beam. While, PVA reinforcement to concrete does not change much the flexural behaviour. With 2% fiber content, the load vs. deflection curves for steel fiber reinforced concrete (SFRC), Steel-PP HFRC and Steel-PVA HFRC are comparatively plotted in Figure 5b. Both S${}_{1.0}$P${}_{1.0}$ and S${}_{1.0}$A${}_{1.0}$ show relatively lower post peak stress history.

### 3.2. Drop Weight Impact Test Results

After drop weight impact tests, the failure patterns of the PC and HFRC discs (rear surface) are shown in Figure 6. As expected, brittle failure occurs to the plain specimen which breaks into halves. On the other hand, HFRC specimen failed mostly by three pieces but they are still connected by bridging fibers crossing the cracks. The hybrid fiber incorporation to concrete may lead to excessive narrow cracks and pulverized matrix while the PC counterpart breaks into separate pieces. This phenomenon may be caused by the stress redistribution in the matrix achieved with the hybrid reinforcement of steel fiber and polymer fiber [18].

#### 3.2.1. Fiber Reinforcement Effect on Impact Response

Table 4 lists all the drop weight impact tests results for concrete with different fiber mixes where increase in the number of post-first crack blow (INPB) is introduced with reference to Rahmani et al. [34]. The $S_{N_{1}}$ is the standard deviation of first crack blows $N_{1}$ while $S_{N_{2}}$ denotes the standard deviation of $N_{2}$ which helps to quantify the amount of variation or dispersion of test data values. For PC, it is interesting to find that the first crack strength ($N_{1}$) and failure strength ($N_{2}$) are the same value of 3 blows which coincides with the experimental results observed by Nia et al. [25]. The reason lies in the fact that the PC specimens fail suddenly through the aggregates in a brittle manner. Meanwhile, the FRC specimens tend to have a much greater failure strength than first crack strength and both $N_{1}$ and $N_{2}$ have an increase to some extent. Therefore, it can be concluded that the fiber reinforcement may contribute to impact toughness, i.e., both first crack strength and failure strength. The toughening enhancement mechanism of fiber on concrete impact resistance mainly stems from the considerable energy absorption during de-bonding, stretching and pullout out of fibers due to the emergence and propagation of cracks in concrete. Once crack occurs, the evenly distributed fibers are activated to arrest the cracking and limit the further crack propagation. Consequently, the strength as well as the ductility of concrete are improved.

The SF, PP and PVA addition effects on impact toughness are evaluated in terms of impact blow times and failure patterns. As indicated by Table 4, the SFRC has the largest failure strength and PP fiber is more effective than PVA fiber which is identical to the flexural strength. Figure 7 gives the bar graphs of FRC impact resistance against the PC counterpart. The impact resistance of SFRC is superior to that of the PC whereby the first crack strength is about 12 times of PC and INPB is improved by 28 blows. Comparing to PP FRC and PVA FRC, the failure strength of SFRC discs show significant increase by 54% and 163%. Therefore, the steel fiber incorporation can better postpone the formation of the first crack and inhibit the crack propagation.

Apart from the impact strength analysis, the effectiveness of the fiber reinforcement can be appreciated from the way the FRC discs failed. The failure pattern in Figure 8a shows that multiple cracking occurs and some cementious matrix pieces are excessively separated from the specimen but still remaining integrity. However, both Figure 8b,c reveals that the concrete discs are broken into two pieces which are accompanied by narrow cracks, small bits of debris and dust without crushing indicating relatively brittle behaviour. Thus, the SFRC failure pattern shows more obvious ductile failure properties under drop impacts.

After mechanical test, the micro structures of transition zone of fibers and cementitious matrix are usually observed via scanning electron microscope (SEM) [10,12]. In Figure 9a, the SF had a smooth surface which was detached from its surrounding matrix. The few hydration products on the SF surface imply the relatively weaker interfacial bond between SF and matrix. The high fiber strength and weaker bond strength make the SF more susceptible to pullout than to rapture. For PP FRC, Figure 9b indicates that PP fibers are encompassed by C-S-H gel and thy are raptured due to the low tensile strength. An obvious difference in the appearance on the fiber surface could be noticed in Figure 9c, where a considerable quantity of hydration products are attached to the PVA fibers indicating the PVA-matrix bond is stronger than the matrix material. The high tensile strength combined with strong bond strength contribute to the Steel-PVA HFRC strength. The fiber surface and matrix of Steel-PVA HFRC after impact were studied in [12].

#### 3.2.2. Hybrid Effect Evaluation on Impact Energy Absorption

The hybrid reinforcement of steel fiber and ploymer fiber may contribute to a better impact energy absorption property since the steel fiber can diffuse more impact energy and polymer fiber delay the cracks extension. In this section, the hybrid effect of Steel-PP and Steel-PVA are discussed with respect to the impact failure energy. Figure 10a shows the impact toughness of Steel-PP HFRC with 2% fiber content. As PP content increases, the first crack strength remains almost constant when PP is less than SF which is then followed by a decrease tend. Meanwhile, the failure strength has the maximum value for the hybrid mixture with 0.5% PP and 1.5% SF. Also, an obvious decrease for $N_{2}$ is observed with PP content increasing ≥1%. It was pointed out by Yap et al. [36] that PP content of PP-Steel HFRC beyond 0.1% is not recommended and only low quantity (≤0.1%) of flexible PP fibers enhances the crack bridging effect. Similarly, we also find that increasing PP content (higher than 0.5%) always negatively affect the impact strength. However, the PVA content effect on the impact toughness is different as shown in Figure 10b whereby the best hybrid mixture corresponding to the largest first crack and failure strength may occur around 1% SF + 1% PVA. This phenomenon is very similar to the energy absorb capacity study results by Zhang and Cao [38] whereby the best hybrid is 1.75% SF + 0.25% PVA.

Figure 11a compares the fiber content effect on the impact toughness with 1% constant SF or PP content. It is interesting to find that to achieve better impact toughness the polymer content is supposed to be between 1% to 1.5% when SF content is 1%. In Figure 11b, both first crack and failure strength increase with the SF content increase from 0.5% to 1.5%. Since the SF has the most effective bridging effect, the SF content may play a more important role in the improvement of impact toughness.

Based on the regression analysis of the impact resistance results, the liner relationship between first crack and failure strength of Steel-PP HFRC and Steel-PVA HFRC. After best fitting, the linear equations describing the first crack and failure strength are developed as $N_{2}=2.78N_{1}+16.3$ for Steel-PP HFRC and $N_{2}=N_{1}+29.6$ for Steel-PVA HFRC as shown in Figure 12. The coefficient of determination ($R^{2}$) is obtained as 0.731 and 0.932 for SP-FRC and SA-FRC fitting equation. According to Ostle [39], a coefficient of $R^{2}$ with 0.7 or higher value is considered as a reasonable model. Therefore, the derived equation may successfully be applied to predict the relationship between the first crack and failure strengths for FRC specimens studied herein.

The hybrid effect of the impact toughness in drop weight impact was evaluated by introducing the hybrid effect index α [8]:(2)$$\alpha =\frac{E_{H}-E_{0}}{\sum \left(E_{i}-E_{0}\right)\beta_{i}}$$
where $\beta_{i}=V_{i}/V$ represents the fiber volume fraction of one kind fiber in the whole volume of fiber *V*, $V_{i}$ is the volume of SF, PP or PVA, $E_{i}$ is the impact toughness of concrete incorporated with single kind fiber, $E_{0}$ is the impact toughness of plain concrete without fiber, $E_{H}$ denotes the impact toughness of HFRC. To exclude the effect of fiber content, the $E_{H}$ calculation should correspond to $E_{i}$ with the same volume fiber reinforcement. In this study, we concentrate on the 2% mixes for further hybrid effect evaluation. If α>1, the hybrid effect is positive for impact toughness improvement, while α<1 means the hybrid effect is negative.

With Equation (Equation 2), the hybrid effect index of Steel-PP HFRC and Steel-PVA HFRC is calculated with respect to the impact energy of drop weight test as shown in Figure 13. The Steel-PP HFRC mixes are featured with hybrid effect index α greater than 1 and 0.67% SF + 1.33% PP has the largest hybrid effect index value. It reveals that the hybridization of SF and PP always plays a positive effect on the impact energy. On the other hand, the hybrid effect index is around 1 for both hybrids with 1.0% SF + 1.0% PVA and 1.5% SF + 0.5% PVA, indicating that impact energy is almost the simply summation of SFRC and PVA FRC. It may be concluded that PVA and SF hybridization does not result in a positive effect on the impact energy unless the PVA fiber volume fraction exceeds the steel fiber volume fraction.

## 4. Conclusions

Hybrid fiber reinforced concrete with PP, PVA and steel fibers was prepared and tested for uniaxial compression, 3-point bending and drop weight tests. The comparative analyses give the conclusions as follows:(1)The improvement of impact energy property of concrete discs can be achieved by the incorporation of polymer (PP/PVA) fiber or steel fiber. The fiber reinforcement transfers the impact failure patterns from brittleness to ductility.(2)Steel fiber addition can better improve the compressive strength, flexural strength and impact strength than its PP or PVA fiber counterpart. Damage modes suggest that the steel fiber tends to pulled out from the matrix while rapture usually occurs to the polymer fiber.(3)A linear equation describing the first crack and failure strength can be best fitted for both Steel-PP HFRC and Steel-PVA HFRC. The derived equation provides a reasonable model for prediction of the relationship of first crack and failure strength.(4)In general, the failure impact energy of Steel-PP HFRC decreases with the increasing PP content while the increasing PVA content may lead to the increase of failure impact energy. With constant 1% PP, the hybrid with more SF cause significant improvement of impact strength.(5)The reinforcement of steel fiber and PP fiber provides positive hybrid effect on improvement of the failure impact energy. For Steel-PP HFRC with 2% fiber content, the impact energy absorption capacity and fiber hybrid effect increase with the increasing of SF dosage.(6)The combined use of steel fiber and PVA fiber suggests less positive hybrid effect on failure impact energy. Especially when theSF content exceeds the PVA content, the hybrid effect index is almost 1, i.e., the corresponding impact energy can be regarded as simply sum of SFRC and PVA FRC.

## Figures and Tables

**Figure 1 materials-11-02563-f001:**
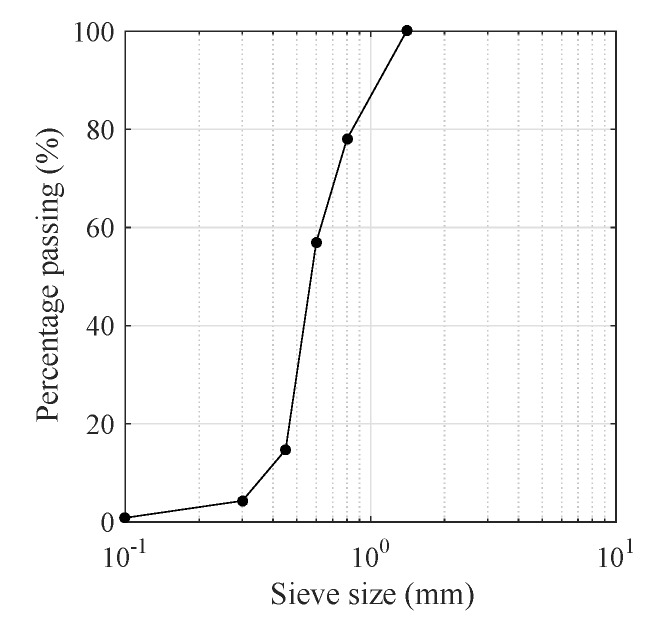
Quartz sand gradation.

**Figure 2 materials-11-02563-f002:**
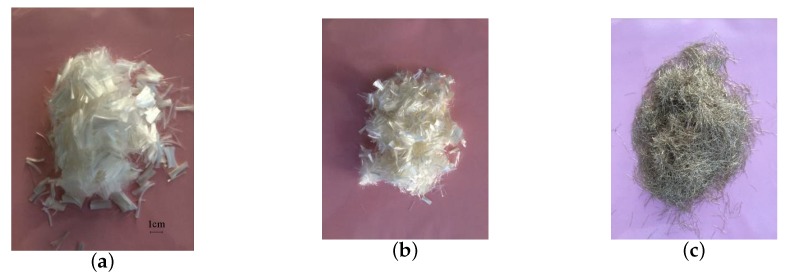
Fiber examples used in this work. (**a**) Polypropylene fiber; (**b**) Polyvinyl alcohol fiber; (**c**) Steel fiber.

**Figure 3 materials-11-02563-f003:**
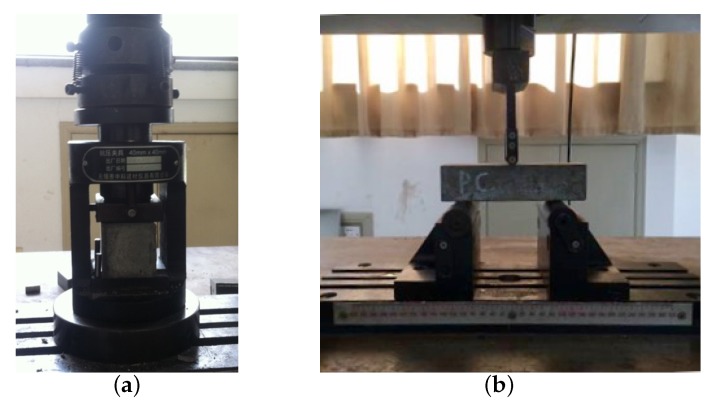
Experimental setup for static test. (**a**) Uniaxial compression test setup; (**b**) 3-point bending test setup.

**Figure 4 materials-11-02563-f004:**
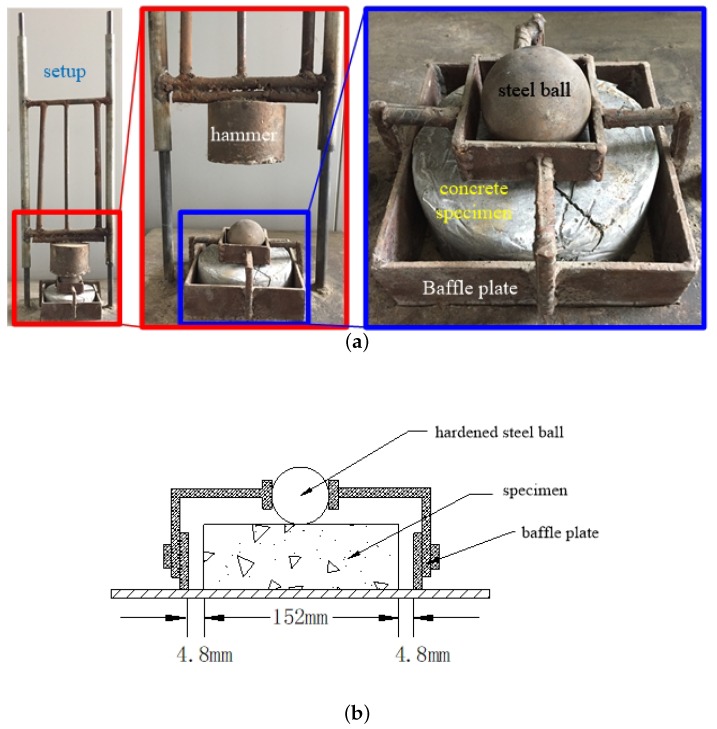
Drop weight test device. (**a**) Setup photo in different views; (**b**) Cross-section view of the disc specimen.

**Figure 5 materials-11-02563-f005:**
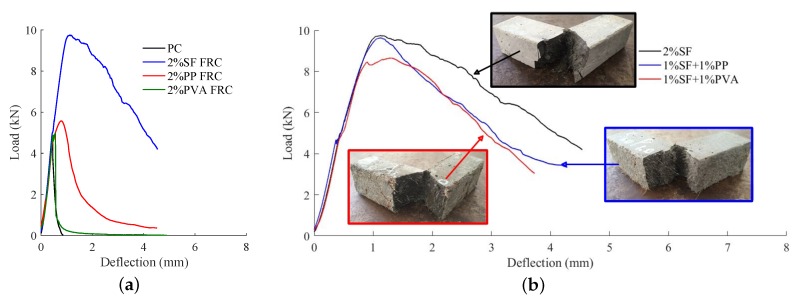
PC and fiber reinforced concrete responses in the 3PBT. (**a**) Load vs. deflection for PC and FRC; (**b**) Bending responses for SFRC and HFRC.

**Figure 6 materials-11-02563-f006:**
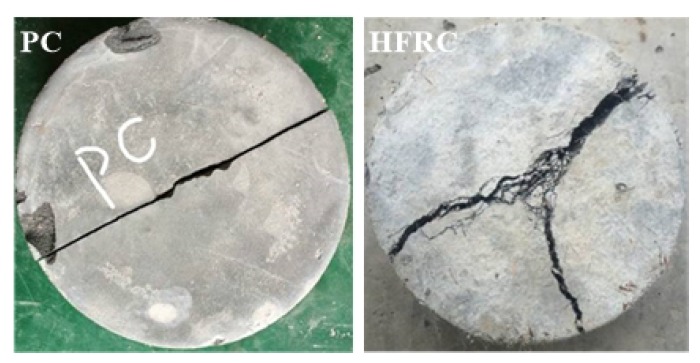
Failure patterns of PC and HFRC specimens.

**Figure 7 materials-11-02563-f007:**
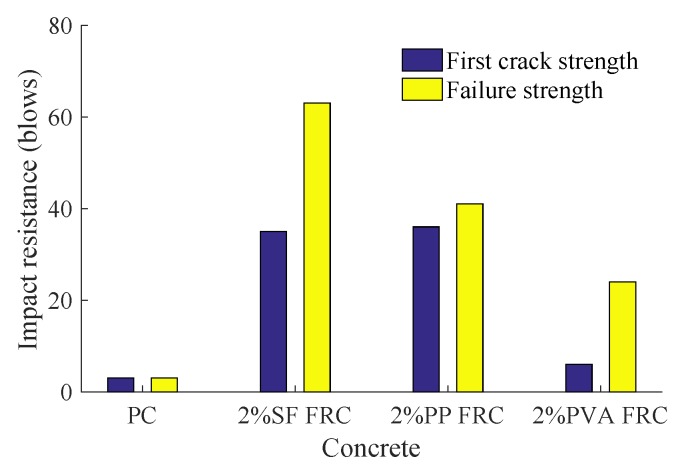
Fiber reinforcement effect on drop impact energy.

**Figure 8 materials-11-02563-f008:**
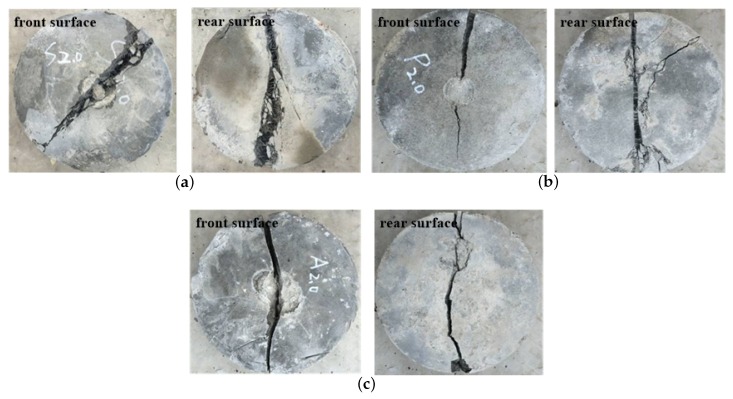
Failure patterns of the FRC specimens after drop weight test. (**a**) FRC with 2% SF; (**b**) FRC with 2% PP; (**c**) FRC with 2% PVA.

**Figure 9 materials-11-02563-f009:**
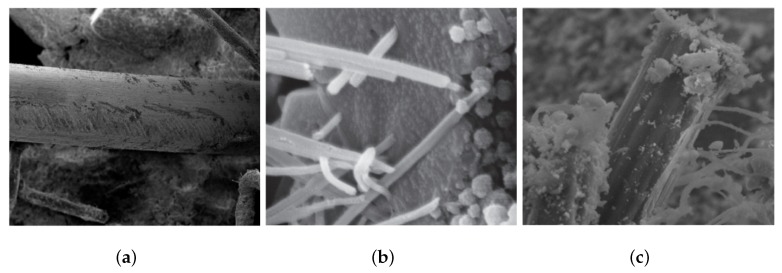
SEM photo of PP, PVA and steel fiber in UHP-HFRC material. (**a**) Steel fiber in matrix [12]; (**b**) PP fiber in matrix [10]; (**c**) PVA fiber in matrix [37].

**Figure 10 materials-11-02563-f010:**
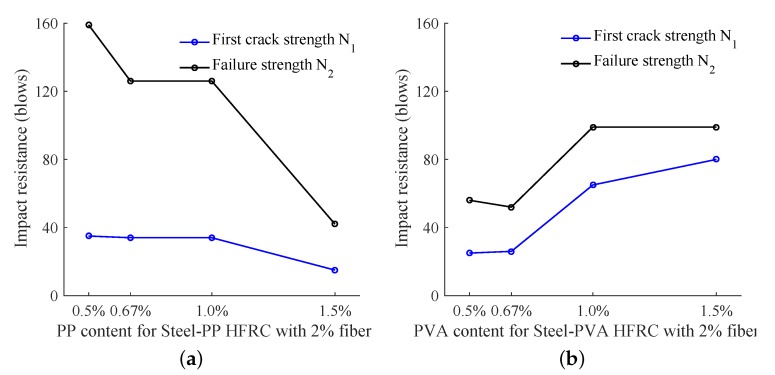
Hybrid effect on drop impact energy. (**a**) Steel-PP HFRC with 2% fiber content; (**b**) Steel-PVA HFRC with 2% fiber content.

**Figure 11 materials-11-02563-f011:**
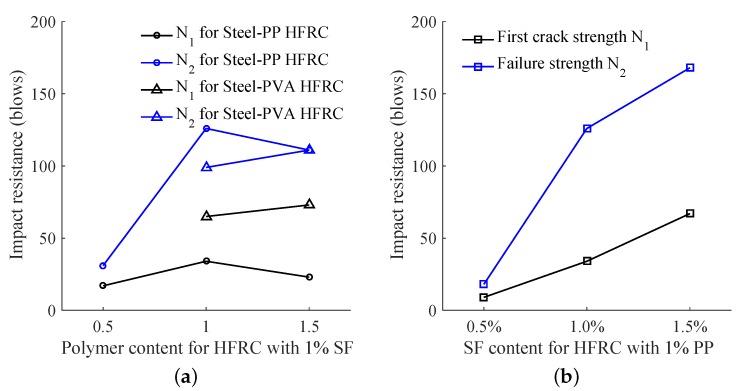
Fiber content effect on drop impact energy. (**a**) HFRC with 1% SF content; (**b**) Steel-PP HFRC with 1% PP content.

**Figure 12 materials-11-02563-f012:**
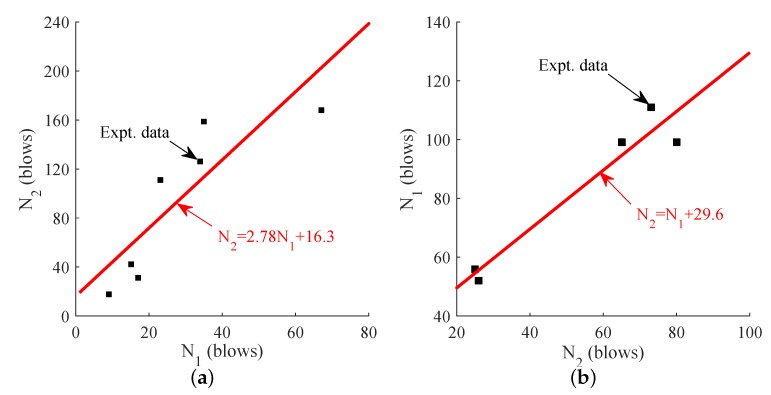
Best fit for first crack and failure strength. (**a**) Steel-PP HFRC impact test results fit; (**b**) Steel-PVA HFRC impact test results fit.

**Figure 13 materials-11-02563-f013:**
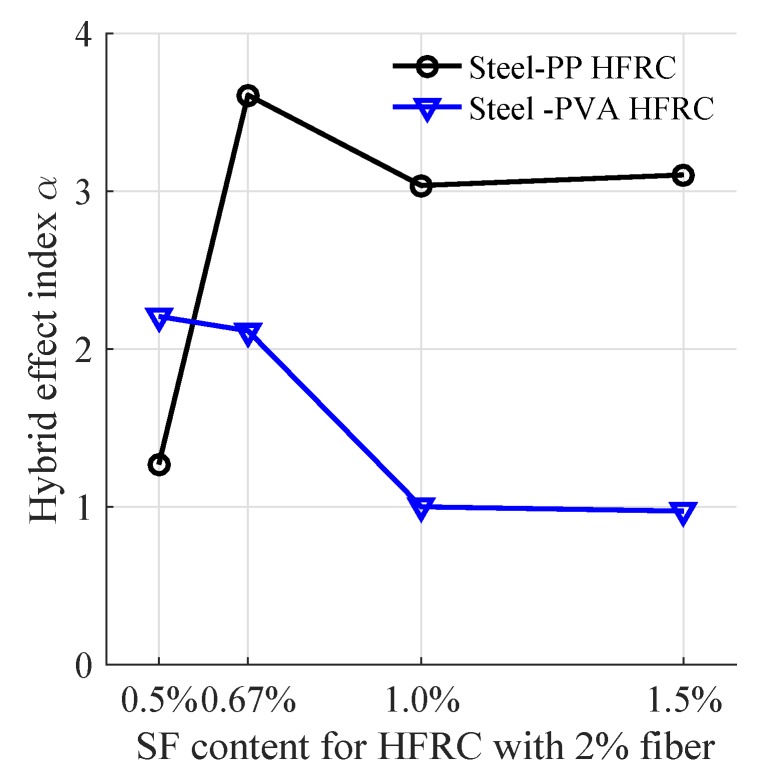
Hybrid effect index in terms of impact energy.

**Table 1 materials-11-02563-t001:** Mixture design of concrete.

Item	Cement	Fly Ash	Water	Quartz Sand	Superplasticizer	Fiber
kg/m${}^{3}$	1165.9	145.7	327.9	590.2	1.2	1.5–2.5% by volume

**Table 2 materials-11-02563-t002:** Fiber information.

Fiber Type	Diameter (μm)	Length (mm)	Density (g/cm${}^{3}$)	Tensile Strength (MPa)	Elastic Modulus (GPa)	Elongation %
PP	30	19	0.91	270	3	4.0–9.0
PVA	26	12	1.30	1000	8	≤40.0
SF	220	12–14	7.85	1200	200	3.5–4.0

**Table 3 materials-11-02563-t003:** Compressive and nominal flexural strength of HFRC.

Mix No.	SF Content	PP Content	PVA Content	7-d $f_{c}$ (MPa)	28-d $f_{c}$ (MPa)	7-d $f_{f}$ (MPa)	28-d $f_{f}$ (MPa)
PC	0	0	0	59.79	70.57	8.67	9.82
S${}_{2.0}$	2.0%	0	0	91.67	115.66	26.95	28.99
P${}_{2.0}$	0	2.0%	0	69.38	90.10	16.18	17.64
A${}_{2.0}$	0	0	2.0%	68.55	84.81	12.34	13.5
S${}_{0.5}$P${}_{1.0}$	0.5%	1.0%	0	65.84	75.26	12.21	13.79
S${}_{0.5}$P${}_{1.5}$	0.5%	1.5%	0	73.84	92.98	16.96	21.21
S${}_{1.0}$P${}_{0.5}$	1.0%	0.5%	0	60.67	85.17	17.03	17.53
S${}_{1.0}$P${}_{1.0}$	1.0%	1.0%	0	69.13	87.24	19.36	21.57
S${}_{1.0}$P${}_{1.5}$	1.0%	1.5%	0	68.77	82.32	17.81	22.85
S${}_{1.33}$P${}_{0.67}$	1.33%	0.67%	0	82.35	105.92	21.22	23.52
S${}_{1.5}$P${}_{0.5}$	1.5%	0.5%	0	84.57	107.76	23.99	28.77
S${}_{1.5}$P${}_{1.0}$	1.5%	1.0%	0	73.93	86.26	23.58	30.84
S${}_{0.5}$A${}_{1.5}$	0.5%	0	1.5%	79.27	95.76	12.85	15.53
S${}_{1.0}$A${}_{1.0}$	1.0%	0	1.0%	78.87	98.35	17.56	19.37
S${}_{1.0}$A${}_{1.5}$	1.0%	0	1.5%	71.85	99.62	18.74	20.45
S${}_{1.33}$A${}_{0.67}$	1.33%	0	0.67%	80.42	105.42	18.55	23.66
S${}_{1.5}$A${}_{0.5}$	1.5%	0	0.5%	83.65	108.55	25.25	28.62

**Table 4 materials-11-02563-t004:** Drop weight impact test results.

Mix No.	First Crack Blows (*N*${}_{1}$)	$S_{N_{1}}$	Failure Blows (*N*${}_{2}$)	$S_{N_{2}}$	INPB
PC	3	0.0	3	0.0	0
S${}_{2.0}$	35	6.6	63	6.6	28
P${}_{2.0}$	36	6.6	41	6.6	5
A${}_{2.0}$	6	6.6	24	6.6	18
S${}_{0.5}$P${}_{1.0}$	9	3.2	18	5.5	9
S${}_{0.5}$P${}_{1.5}$	15	5.9	42	14.8	27
S${}_{1.0}$P${}_{0.5}$	17	1.4	31	3.3	14
S${}_{1.0}$P${}_{1.0}$	34	5.1	126	30.5	92
S${}_{1.0}$P${}_{1.5}$	23	4.5	95	14.5	72
S${}_{1.33}$P${}_{0.67}$	34	6.6	126	41.7	92
S${}_{1.5}$P${}_{0.5}$	35	3.5	159	52.2	164
S${}_{1.5}$P${}_{1.0}$	67	7.4	168	45.0	101
S${}_{0.5}$A${}_{1.5}$	80	18.9	99	36.5	24
S${}_{1.0}$A${}_{1.0}$	65	8.1	99	16.5	35
S${}_{1.0}$A${}_{1.5}$	73	15.3	111	27.5	36
S${}_{1.33}$A${}_{0.67}$	26	5.5	52	12.1	26
S${}_{1.5}$A${}_{0.5}$	25	7.9	56	22.6	31

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
