# Peer review of "Experimental Study on Hybrid Effect Evaluation of Fiber Reinforced Concrete Subjected to Drop Weight Impacts"

_materials, 2018, doi:10.3390/ma11122563_

Round 1
Reviewer 1 Report
This paper presents an experimental study to investigate the hybrid effect of different fibers on the impact toughness of hybrid fiber reinforced concrete (HFRC) under drop weight tests. The study also exhibits the compressive and flexural strength at 7 and 28 days of curing. Steel, PP and PVA fibers were used to produce the HFRC mixtures. The hybrid effect of mixes of steel-PP and steel-PVA fibers on the impact toughness was evaluated.
Some suggestions to enhance the paper quality are as follows:
Page 4, lines 51-52: Correct the sentence in these lines.
Section 2.1:
- How was the slump for all mixtures?
- The authors should provide the aspect ratio of the fibers. Is this parameter having effect on the impact toughness of HFRC?
Section 2.3:
- The number of blows to cause the failure strength (N2) was defined as the number of blows to spread the cracks sufficiently. When it was considered that the failure was sufficient for FRC?
Section 3:
- Could the specimens small sizes affect the results obtained in compressive and flexural tensile strength tests? Please discuss briefly.
- Can the authors give more interpretation of the found results with comparison of the literature results?
- Why the impact toughness was not calculated in this paper?
- Figure 7 could be omitted.
- Figures 8, 11, 12, 13 - The number of blows is not the strength. Please correct the graphs.
- Why the relationships of impact toughness and fiber content are not presented in the graphs?
Author Response
Many thanks for your careful and valuable comments. We have made revisions according to your request.
Reviewer 2 Report
The authors present an interesting study on the effect of multi-fiber reinforcement in concrete by mixing long steel fibers with short PP and PVA, on the impact fracture properties of the resulting material. The research methodology is correct, but the main concern is that the comparison between both types of fibers might be wrong provided that different geometrical properties (affecting to the bonding behavior of the fiber-cement interface) have been accounted for. Apart from that, the paper contains serious grammatical flaws and a thorough language revision should be carried out. For this reason, the authors are kindly requested to discuss on this issue before making a decision on the publication of the paper. In other words, if such difference in the bonding behavior is proved right, the conclusions presented in the article might be wrong. Finally, the following comments/suggestions are provided:
The introduction is correct, complete and very well structured. The objectives are also clear.
Line 8: Fiber also help arrest micro cracks before the peak, therefore, during the hardening behavior of the composite. Please remark this.
Line 12: use "multiple phases" instead of "multi-phases".
Line 49: The term "fraction properties" might be wrong, please use "mixing contents" or something similar.
Line 74: use "evenly" or "homogeneously" distributed, rather than "randomly".
Line 90: remove "corporation".
Figure 5: Why does the PP reinforced specimen present more strength and ductility than that of PVA? According to the fiber properties, it should be the opposite, as long as the bonding surface are equivalent. If not, the comparison would be meaningless.
Author Response

(The authors gave the same response as above.)

Reviewer 3 Report
The paper concerns the investigations about the hybrid effect of several types of fibers on mechanical performance of concrete. The study presented is very interesting and presents actual topic in terms of main research directions in concrete technology. Authors tested a lot of concrete compositions giving big comparative possibilities and analysis. Experiment is well designed and described properly. The results obtained have very high scientific value, but paper requires major revision before acceptance for publication. Detailed comments are listed below:
1. There are few language, syntax errors. I suggest checking the article by English native speaker.
2. Formatting is not compatible with the Materials template. Please correct this.
3. Introduction - the authors wrote that each level of material can be strengthened by fibers, in the scale of coarse aggregate – steel fibers can be used, in the scale of fine aggregate – PP or PVA, in scale of cement grains – e.g., the carbon nanotubes – this information should be added, an example reference:
Szeląg M.: Mechano-physical properties and microstructure of carbon nanotube reinforced cement paste after thermal load. Nanomaterials, vol. 7(9), 2017, s. 267
4. Introduction - please also provide information that as dispersed reinforcement some natural fibers can be used, e.g., the flax fibers, an exemplary reference to literature:
Brzyski P., Barnat-Hunek D., Fic S., Szeląg M.: Hydrophobization of lime composites with lignocellulosic raw materials from flax. Journal of Natural Fibers, vol. 14(5), 2017, s. 609-620
5. Table 1 should be placed below second chapter.
6. Figure 2 - please include the scale in the figure so that the row of the size of the fibers can be seen.
7. Section 2.2. - please state if the samples were maturated in water or in a climatic chamber in which there was very high humidity.
8. Table 3 and 4 - please give some basic measure of the results distribution, e.g., standard deviation or coefficient of variation; there is an inaccuracy between the Mix No. and the SF, PP, PVA content columns, please correct this.
9. Information about the amount of samples tested should be placed in the test methods section, similarly information contained in the 118th row (formula for calculating the nominal flexural strength).
10. Information about the microstructure test using SEM should be placed in the test methods section. Please also describe how the samples were prepared for the test.
Author Response
Many thanks for your careful and valuable comments. We have made revisions according to your request.

Reviewer3: The paper concerns the investigations about the hybrid effect of several types of fibers on mechanical performance of concrete. The study presented is very interesting and presents actual topic in terms of main research directions in concrete technology. Authors tested a lot of concrete compositions giving big comparative possibilities and analysis. Experiment is well designed and described properly. The results obtained have very high scientific value, but paper requires major revision before acceptance for publication. Detailed comments are listed below:
Many thanks for your careful and valuable comments. We have made revisions according to your request, please check the below response.
1.There are few language, syntax errors. I suggest checking the article by English native speaker.
Thank you for your suggestion. We have revised some language errors, and hope it reads more smoothly to you now.
2.Formatting is not compatible with the Materials template. Please correct this.
Thank you.
We used the Latex to write our manuscript. The template of Latex manuscript for this journal is this format. We hope it makes sense to you.
3. Introduction - the authors wrote that each level of material can be strengthened by fibers, in the scale of coarse aggregate – steel fibers can be used, in the scale of fine aggregate – PP or PVA, in scale of cement grains – e.g., the carbon nanotubes – this information should be added, an example reference:
Szeląg M.: Mechano-physical properties and microstructure of carbon nanotube reinforced cement paste after thermal load. Nanomaterials, vol. 7(9), 2017, s. 267
4. Introduction - please also provide information that as dispersed reinforcement some natural fibers can be used, e.g., the flax fibers, an exemplary reference to literature:
Brzyski P., Barnat-Hunek D., Fic S., Szeląg M.: Hydrophobization of lime composites with lignocellulosic raw materials from flax. Journal of Natural Fibers, vol. 14(5), 2017, s. 609-620
Thank you! We have revised the manuscript according to your request. And these two literatures have been cited in the revised version as Ref. [9] and [15]. Please check!
5. Table 1 should be placed below second chapter.
Thank you for your suggestion. We have revised accordingly.
6.Figure 2 - please include the scale in the figure so that the row of the size of the fibers can be seen.
We agree with you!
Please check the updated manuscript.
7.Section 2.2. - please state if the samples were maturated in water or in a climatic chamber in which there was very high humidity.
Thanks for your comment. We also believe it is important to include this information.
So we added at lines 74-75:’In the climatic chamber with 90% humidity, the FRC samples were prepared with the following mixing procedures.’
Please check!
8. Table 3 and 4 - please give some basic measure of the results distribution, e.g., standard deviation or coefficient of variation; there is an inaccuracy between the Mix No. and the SF, PP, PVA content columns, please correct this.
Thank you very much. It was our bad for the incorrect Mix No. We have corrected them.
It is import to include the standard deviation to the data in the test for the readers to better understand and analyze the data.
At lines 156-157, it was added that: ‘The SN1 is the standard deviation of first crack blows N1 while SN2 denotes the standard deviation of N2 which helps to quantify the amount of variation or dispersion of test data values.’ Please check!
9. Information about the amount of samples tested should be placed in the test methods section, similarly information contained in the 118th row (formula for calculating the nominal flexural strength).
Thanks! We have revised accordingly
10. Information about the microstructure test using SEM should be placed in the test methods section. Please also describe how the samples were prepared for the test.
Thanks! But we are sorry that these SEM figures are from literature. We didn’t take the SEM photo to better analyze the microstructure of the mixtures which should be done in the future work.
Round 2
Reviewer 3 Report
All comments have been included. I accept the paper for publication.